# FedEED: Efficient Federated Distillation with Ensemble of Aggregated Models

## Abstract

In this paper, we study the key components of distillation-based model aggregation in federated learning (FL). For that purpose, we first propose a generalized distillation framework which divides the training and model aggregation process into three key stages and includes existing methods as special cases. By investigating the contribution of each stage, we propose a novel distillation-based FL scheme, named Federated Efficient Ensemble Distillation (FedEED). Different from existing approaches, the ensemble teacher of FedEED is constructed by aggregated models, instead of the client models, to achieve improved scalability in large-scale systems. Due to the use of aggregated models, FedEED also achieves higher level of privacy protection, because the access to client models is no longer required. Furthermore, the knowledge distillation in FedEED only happens from the ensemble teacher to a designated model such that the diversity among different aggregated models is maintained to improve the performance of the ensemble teacher. Experiment results show that FedEED outperforms the state-of-the-art FL schemes, including FedAvg and FedDF, on the benchmark datasets. Besides the performance advantage, the designated distillation also allows for parallelism between server-side distillation and clients-side local training, which could speed up the training of real world systems.

## 1 Introduction

Federated learning (FL) (McMahan et al., 2017) allows users to jointly train a deep learning model without sharing their own data. Recent works (Lin et al., 2020; Zhang et al., 2022; Huang et al., 2022; Cho et al., 2022) adopted knowledge distillation (Hinton et al., 2015) for model aggregation to tackle data and device heterogeneity issues. These distillation-based aggregation methods have been shown to outperform weight averaging methods like FedAvg (McMahan et al., 2017). However, existing works utilized all client models to build the ensemble for knowledge distillation, leading to poor scalability in real world applications with a large number, e.g., thousands, of clients. Furthermore, existing methods built the ensemble teacher by similar client models, where the low diversity among the compositing models will limit the performance of the ensemble teacher.

To tackle the above issues, we will study the key components of distillation-based FL and propose an efficient and scalable distillation method. For that purpose, we first introduce a generalized framework for distillation-based model aggregation, which consists of three major components, namely the local trainers, the ensemble trainer, and the global trainer. The local trainers perform local training on the clients-side, where each local trainer handles a sub-group of participating clients. After that, each local trainer collects and aggregates the updated client models, and constructs one global model. Note that this is the global model for all participating clients in a sub-group. Existing FL algorithms like FedAvg (McMahan et al., 2017) can be used in-place as a local trainer. Then, the ensemble trainers combine the updated global models and local models into an ensemble, providing a higher capacity than the stand-alone global models. Finally, the global trainer will utilize the ensemble to further enhance the global models via knowledge distillation.

With the generalized framework, we investigate the contributions of each key component. In particular, we compare the performance of the distillation-based FL with different local, ensemble, and global trainers and try to tackle the limitations of the existing methods in a bottom-up manner. We focus on several aspects of the distillation framework including: 1.) Improving scalability and pri-

vacy of distillation-based aggregation by building the ensemble teacher from a set of aggregated models, i.e., client models are no longer required to construct the ensemble teacher; 2.) Maximizing the capacity of the ensemble and global models by maintaining the diversity among models utilized to create the ensemble; 3.) Reducing the computation overhead by exploiting the parallelism between server-side and client-side training. By integrating the above ideas together, we propose a new algorithm named Federated Efficient Ensemble Distillation (FedEED). FedEED is a highly scalable, distillation-based FL algorithm that does not require direct access to the client models, which provides further protection to the user privacy. The proposed FedEED achieved state-of-the-art results in CIFAR10/100 (Krizhevsky et al., 2009) with the non-independent and identically distributed (Non-IID) data.

The contributions of this paper include:

1) We propose a generalized framework for distillaion-based model aggregation, which can be viewed as a generalization of the existing distillation-based FL algorithms. With the proposed framework, the contribution of each key component can be investigated to build up more efficient algorithms.

2) By investigating the contribution of each component in the generalized framework, we propose FedEED, a highly efficient and scalable, distillation-based FL algorithm with improved privacy and model diversity. Experiment results demonstrate that FedEED can achieve the state-of-the-art performance with lower complexity and latency than existing federated distillation algorithms.

## 2 RELATED WORKS

**Federated learning.** Deep learning has obtained great successes in the last decade. However, in practice, large amount of user data can not be shared to the central server due to privacy regulations and communication constraints. To tackle the above issues, federated learning (McMahan et al., 2017) was proposed to train a global model based on data belonging to different users, without data sharing. The simplest approach is FedAvg (McMahan et al., 2017), which performs multiple local epochs of training on the client-side, and then aggregates the updated client models with weight averaging. Other approaches, such as FedProx (Li et al., 2020), also utilize weight averaging to perform model aggregation, but with added regularization to tackle data heterogeneity issues. In this paper, we focus on distillation-based model aggregation for FL. For both the generalized framework and the newly proposed FedEED, averaging based methods such as FedAvg and FedProx can be directly applied as the local trainers. This makes the proposed framework and FedEED compatible with different weight averaging methods.

**Knowledge distillation.** Knowledge distillation (Hinton et al., 2015) has been proposed in deep learning to compress deep neural networks. Student models, which typically have a smaller size, are forced to mimic the output of the teacher model. In some recent works, knowledge distillation has been applied in FL. There are two types of approaches to apply knowledge distillation in FL. The first type utilizes distillation to perform model aggregation. For example, FedDF (Lin et al., 2020) and FedBE (Chen & Chao, 2021) utilized client models as a teacher to update the global model on the server, and the purpose is to improve the model aggregation performance with heterogeneous data. Some other works, including FedFTG (Zhang et al., 2022) and Fed-ET (Cho et al., 2022), utilized distillation for the same purpose, but in a different setting, i.e. in a data-free or model heterogeneous environment. The second type shares model predictions between clients and the server for training purposes. For example, in FD (Jeong et al., 2018), model predictions are shared between clients to regularize the local training. In FedAD (Gong et al., 2021), model outputs on the client-side are used to train a global model on the server through distillation.

The proposed generalized framework and FedEED in this paper can be classified into the first type. In fact, the framework is a generalization of the available works in the first type. However, the mechanisms FedEED utilizes to improve its scalability, privacy and performance are orthogonal to those of the existing works, and can be combined with existing methods to further improve the performance.

---

**Algorithm 1** Generalized Framework for Model Aggregation

---

**function** LOCALTRAINER($w_0^l, ..., w_{N-1}^l$)
    // Update local models $w_0^{l*}, ..., w_{N-1}^{l*}$
    // Update global model $w^g$
    **return** $w^g, w_0^{l*}, ..., w_{N-1}^{l*}$
**end function**
**function** ENSEMBLETRAINER($w_0^g, ..., w_{K-1}^g, w_{0,0}^l, ..., w_{K-1,N_{K-1}-1}^l$)
    // Update ensemble $w^{e*}$
    **return** $w^{e*}$
**end function**
**function** GLOBALTRAINER($w^{e*}, w_0^g, ..., w_{K-1}^g$)
    // Update global models $w_0^{g*}, ..., w_{K-1}^{g*}$
    **return** $w_0^{g*}, ..., w_{K-1}^{g*}$
**end function**
// Initialize $K$ global models $w_{-1,0}^{g*}, ..., w_{-1,K-1}^{g*}$
**for** $t \in \{0, ..., T-1\}$ **do**
    // Split clients into sets $S_{t,0}, ..., S_{t,K-1}$ with sizes $N_{t,0}, ..., N_{t,K-1}$
    **for** $k \in \{0, ..., K-1\}$ **do in parallel**
        $w_{t,k,0}^l, ..., w_{t,k,N_{t,k}-1}^l \leftarrow w_{t-1,k}^{g*}$
        $w_{t,k}^g, w_{t,k,0}^{l*}, ..., w_{t,k,N_{t,k}-1}^{l*} \leftarrow LocalTrainer(w_{t,k,0}^l, ..., w_{t,k,N_{t,k}-1}^l)$
    **end for**
    $w_t^{e*} \leftarrow EnsembleTrainer(w_{t,0}^g, ..., w_{t,K-1}^g, w_{t,0,0}^{l*}, ..., w_{t,K-1,N_{t,K-1}-1}^{l*})$
    $w_{t,0}^{g*}, ..., w_{t,K-1}^{g*} \leftarrow GlobalTrainer(w_t^{e*}, w_{t,0}^g, ..., w_{t,K-1}^g)$
**end for**

---

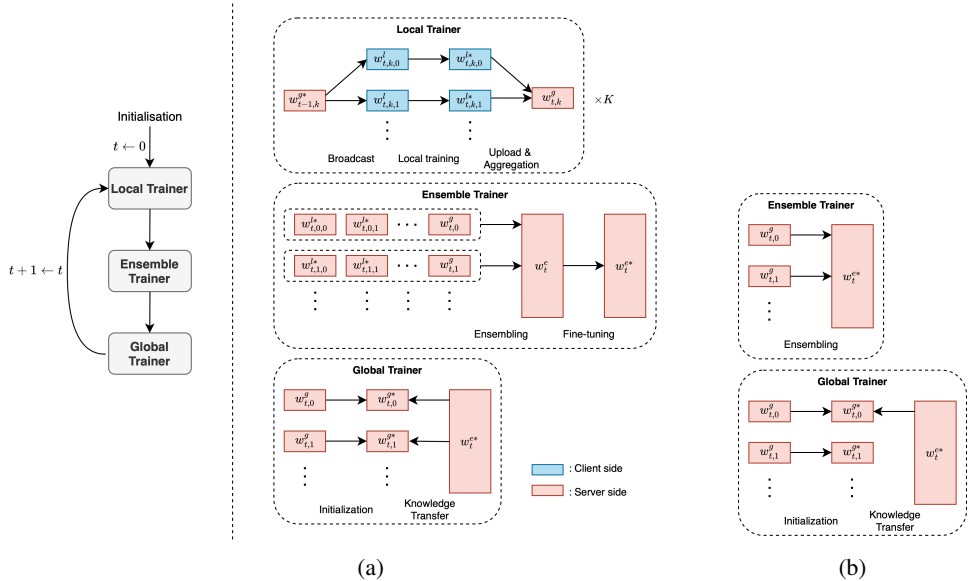

(a)          (b)

Figure 1: (a) Overview of the generalized framework for model aggregation. The number of local trainer, the implementation of local training, ensembling, etc., are subject to the implementation. (b) The Ensemble trainer and global trainer of FedEED. There are two key differences. First, the ensemble trainer in FedEED only utilizes the $K$ global models to build the ensemble. Second, the global trainer only update one of the global models by distillation.

## 3   THE GENERALIZED FRAMEWORK FOR MODEL AGGREGATION

To investigate the potential of utilizing distillation for model aggregation in FL, we propose the generalized framework, which includes $K$ local trainers, an ensemble trainer, and a global trainer, as shown in Fig. 1a. Each trainer can be replaced by different implementations to form different algorithms. Note that the generalized framework keeps and updates $K$ global models simultaneously, which includes the existing methods as a special case when $K = 1$. In every training round, each local trainer will distribute one of the global models to a subset of clients, perform local training, and aggregate the trained client models into a updated global model. Then, the ensemble trainer

will aggregate the trained client models and/or the updated global models into an ensemble model. Finally, the global trainer will compress the knowledge of the ensemble and further enhance the global models.

Many existing algorithms can be viewed as a special case of the generalized framework. For example, by utilizing FedAvg (McMahan et al., 2017) as the local trainer, combining all client models as the ensemble, and implementing the conventional knowledge distillation (Hinton et al., 2015), FedDF (Lin et al., 2020) can be realized. By applying multiple local trainers, each of which maintains a global model with different architectures or hyper-parameters, the heterogeneous learning setting (Huang et al., 2022) can be implemented.

By generalizing existing algorithms, the framework makes the design and implementation of FL systems easier, where different components can be updated according to different design objectives. Furthermore, knowledge distillation (Hinton et al., 2015) is still an active research topic, and the generalized framework allows easier integration of different distillation methods into FL. In the following, we introduce the components of the generalized framework, i.e. the local trainers, the ensemble trainer and the global trainer. The overall workflow of the framework is illustrated in Algo. 1 and Fig. 1a.

### 3.1 LOCAL TRAINER

In each communication round, all available clients will be partitioned into $K$ groups, where each group of clients will be handled by one local trainer. For example, in the $t$-th round, the task of the $k$-th local trainer is to update the $k$-th global model by performing local training and model aggregation over the $k$-th group of clients. In particular, the local trainer first distributes the weight of the global model in the previous round, i.e., $w_{t-1,k}^{g*}$, to its clients to initialize the client model weights $w_{t,k,0}^l, ..., w_{t,k,N_{t,k}-1}^l$, where $w_{t,k,n}^l$ denotes the local model of the $n$-th client in the group with $0 <= n < N_{t,k}$, $N_{t,k}$ denotes the number of clients associated with the $k$-th local trainer in the $t$-th round. Then, the local trainer performs training on the client-side with the clients' private dataset, and updates the model of the $n$-th client as $w_{t,k,n}^{l*}$. Lastly, the local trainer collects and aggregates model weights from its clients to the new global model $w_{t,k}^g$. Note that different local trainers can be associated with models with different architectures, which allows the framework to learn heterogeneous models. In this paper, we only consider the scheme which we randomly and evenly map the clients to the local trainers in each training round, such that there are $\frac{C}{K}$ clients per group, where $C$ is the number of active clients.

Many existing FL algorithms can be directly utilized as a local trainer, e.g. FedAvg (McMahan et al., 2017), and each local trainer can be a different learning algorithm. Moreover, learning algorithms with local regularization can be used in-place to tackle the Non-IID issue, e.g. FedProx (Li et al., 2020) and SCCAFOLD (Karimireddy et al., 2020). In this work, we consider three options for the local trainer, including FedAvg, FedProx and SCCAFOLD. We expect the local trainer to have relatively simple relation with the higher level trainers, such that faster local training can speed up the training of the ensemble and hence, the global model. We will evaluate the compatibility of these trainers with different higher level trainers.

### 3.2 ENSEMBLE TRAINER

After updating the global models by the local trainers, the ensemble trainer constructs the ensemble teacher model (Hinton et al., 2015) with weight $w_t^e$ by the the client models and/or global models obtained by the local trainers. The ensemble trainer may also fine-tune the ensemble weight to $w_t^{e*}$ with a server dataset. Composed by multiple models, the ensemble typically has better performance than the individual models. It can serve as the teacher model for knowledge distillation in the later stage, and also be directly utilized for inference on the server or some high power devices.

There are multiple ways to build an ensemble. For example, FedDF (Lin et al., 2020) utilized all client models in a round to construct the ensemble; FedBE (Chen & Chao, 2021) sampled new models from the distribution built from client models and utilized both the sampled models and client models to construct the ensemble for a higher capacity. Unlike the above works which utilized knowledge distillation to compress a single and compact model, Fed-ensemble (Shi et al., 2021) learned multiple global models and utilized them to build the ensemble for inference.

## 3.3 GLOBAL TRAINER

At the last stage of each communication round, the ensemble obtained from the ensemble trainer is utilized as the teacher model for knowledge distillation to enhance the global models. We update the weights of the global models from $w_{t,0}^g, ..., w_{t,K-1}^g$ to $w_{t,0}^{g*}, ..., w_{t,K-1}^{g*}$, by mimicking the ensemble teacher's outputs with some sample inputs. For example, we can minimize the KL-divergence to enhance the global models for a classification task (Hinton et al., 2015). Existing works have shown that distillation can be utilized to perform model aggregation in FL. For instances, FedDF (Lin et al., 2020) utilized a basic knowledge distillation scheme in each round of training. On the other hand, FedBE (Chen & Chao, 2021) applied SWA (Izmailov et al., 2018) to prevent the global model from being over-fitted with the noisy teacher output.

## 4 EXPERIMENTS: THE GENERALIZED FRAMEWORK

In this section, we study the effect of the ensemble trainer and the global trainer.

### 4.1 EXPERIMENTS: ENSEMBLE TRAINER

There are three critical issues in constructing the ensemble trainers, namely privacy, scalability, and model diversity. Existing distillation-based aggregation methods (Lin et al., 2020; Zhang et al., 2022; Huang et al., 2022; Cho et al., 2022) require all client models to build the ensemble, which cause privacy concerns since the models can leak client's information. See section A.6 in the appendix for more details. Furthermore, their scalability is limited, because the cost of inference with the ensemble is proportional to the number of compositing models and the server may not be able to efficiently perform the training if the number client models is too large. Existing methods didn't address this issue or assume that the computation cost to the server is insignificant comparing to the communication time Chen & Chao (2021). With FedDF (Lin et al., 2020) and FedBE (Chen & Chao, 2021), the diversity between the models in the ensemble is determined by the local training of a single round and may be limited, reducing the capacity of the ensemble.

Fed-ensemble (Shi et al., 2021) trains multiple global models and builds the ensemble from these global models for inference. The ensemble does not scale with the number of clients, but scales with the number of global models $K$, which is a hyper-parameter. Moreover, the diversity of the models constructing the ensemble is maintained across the whole training, because different global models experience different training sequences. The idea can be utilized to build the ensemble for distillation purposes, which allows for different ensemble complexity, determined by the number of global models $K$. However, increasing the capacity of the ensemble by increasing $K$ will also reduce the number of clients per global model, and thus slows down the convergence. We thus propose to utilize the last $R$ checkpoints of each of the global models to build the ensemble, which increases the capacity of the ensemble without slowing down the training of the individual global models.

In this paper, we consider five ways to build the ensemble:

1) Vanilla Ensemble. Similar to FedDF (Lin et al., 2020), we utilize all client models to build the ensemble and only train one global model (i.e. $K = 1$). For an input $x$, we denote the ensemble function by $F$, where $F(x|w_0, ..., w_{N-1})$ is the class probabilities with averaged logits of the networks that are weighted by $w_0, ..., w_{N-1}$. We compute the output $y$ by $y = F(x|w_{t,0,0}^{l*}, ..., w_{t,0,N_0-1}^{l*})$.

2) Bayesian Ensemble. Following FedBE (Chen & Chao, 2021), we train a global model and build ensemble by combining the client models, the averaged model, and $M$ sampled models $w_{t,m}^s$ with $0 <= m < M$, either from the Gaussian distribution or the Dirichlet distribution. The ensemble can be expressed as $y = F(x|w_{t,0,0}^{l*}, ..., w_{t,0,N_0-1}^{l*}, w_{t,0}^g, w_{t,0}^s, ..., w_{t,M-1}^s)$.

3) Ensemble from multiple global models. Like Fed-ensemble (Shi et al., 2021), we train $K$ global models, each with different initialization seeds and local training sequences, and combine them into an ensemble with $y = F(x|w_{t,0}^g, ..., w_{t,K-1}^g)$.

4) Ensemble from client models, which are initialized from different global models (Lin et al., 2020; Shi et al., 2021). This scheme is similar to 3), except that we utilize the client models to build the ensemble with $y = F(x|w_{t,0}^{l*}, ..., w_{t,K-1,N_{K-1}-1}^{l*})$.

Table 1: CIFAR10 results of the generalized framework with different ensemble trainers.

| Model | Method | CIFAR10 | |
|---|---|---|---|
| | | $\alpha = 1.0$ | $\alpha = 0.1$ |
| ResNet20 | Global model ($K = 1$) | $88.53 \pm 0.31$ | $78.72 \pm 2.31$ |
| | Ensemble ($K = 1$, Clients) | $88.82 \pm 0.21$ | $80.22 \pm 0.88$ |
| | Ensemble ($K = 1$, Clients, Weighted) | $88.79 \pm 0.16$ | $80.11 \pm 0.95$ |
| | Ensemble ($K = 1$, Bayesian, Gaussian) | $88.72 \pm 0.14$ | $79.60 \pm 1.26$ |
| | Ensemble ($K = 1$, Bayesian, Dirichlet) | $88.70 \pm 0.11$ | $80.25 \pm 1.43$ |
| | Global model ($K = 4$) | $86.69 \pm 0.54$ | $70.74 \pm 5.11$ |
| | Ensemble ($K = 4$, Aggregated) | $90.58 \pm 0.31$ | $81.76 \pm 2.40$ |
| | Ensemble ($K = 4$, Clients) | $90.49 \pm 0.17$ | $82.12 \pm 1.65$ |
| | Ensemble ($K = 4$, $R = 2$, Aggregated) | $\underline{90.75 \pm 0.24}$ | $83.50 \pm 1.56$ |
| | Ensemble ($K = 4$, $R = 4$, Aggregated) | $90.69 \pm 0.19$ | $83.99 \pm 0.88$ |

5) Ensemble from multiple aggregated global models in multiple rounds. In addition to scheme 3), we use the global models from the last $R$ checkpoints to build the ensemble with $y = F(x|w_{t,0}^g, ..., w_{t,K-1}^g, ..., w_{t-R+1,0}^g, ..., w_{t-R+1,K-1}^g)$.

We compare the performance of the above ensemble strategies on CIFAR10 dataset with Non-IID settings. We performed two runs, where we trained one ($K = 1$) and four ($K = 4$) global models with the generalized framework, respectively. We used FedAvg (McMahan et al., 2017) as the local trainer, and to better illustrate the effect of the ensemble trainer, we skipped the global trainer, i.e. we did not perform distillation. For the run with $K = 1$, we built 1) vanilla ensembles, including a variant which is weighted by the number of training samples held by the client, and 2) Bayesian ensembles (both Gaussian and Dirichlet). We also performed similar evaluation for the runs with $K = 4$, where we built 3) ensembles from global models of one round (i.e. $R = 1$), 4) ensembles form client models, and 5) ensembles from multiple checkpoints of the global models (with $R = 2$ or $R = 4$). For the details of the experiment setting, see Sec. 6.

The results are shown in Table 1, and also Fig. 2 in the appendix. With low degree of Non-IID ($\alpha = 1.0$), despite slowing down the convergence of the individual global model, increasing the number of global models provides the best results. All ensembles with $K = 4$ perform closely, where the ensembles built by global models from multiple checkpoints provide slightly higher accuracy. With $K = 1$, all methods do not show clear performance advantage over the global model, which agrees with our understanding about the diversity among the compositing models.

With highly Non-IID data, all ensemble strategies showed significant improvement over the global model with $K = 1$. However, with only global models of a single round, the ensemble with $K = 4$ outperformed the strategies with $K = 1$, which requires access to the client models. By increasing $R$ from 1 to 2 or 4, we observed substantial performance improvements by the ensemble with $K = 4$, outperforming all methods with $K = 1$ by a large margin. Moreover, with $K = 4$ and $R = 1$, the ensembles built from global models and client models perform closely. It indicates that accessing client models is not necessary to build a high performance ensemble.

In the experiment, ensembles built by following option 5) achieved the best performance, with better privacy protection and higher scalability. Thus, we consider option 5) the candidate ensemble trainer.

## 4.2 EXPERIMENTS: GLOBAL TRAINER

In Sec. 4.1, it was shown that the best performance can be achieved by learning multiple global models and using multiple check-points of these models. However, if we are interested in a single and compact model for inference, we need to compress the ensemble into a global model. In the cases with multiple global models in the generalized framework, there is no clear rule for distillation, and we consider multiple distillation schemes here.

Firstly, we consider a basic knowledge distillation scheme, i.e. we utilize the ensemble as the teacher and all global models are trained by mimicking the output of the ensemble. Note that we fix the weight of the ensemble during the distillation process. This is similar to the heterogeneous version of FedDF (Lin et al., 2020), where all global models will be improved by knowledge distillation.

Table 2: CIFAR10 results of the generalized framework with different global trainers.

| Model | Method | CIFAR10 | |
| --- | --- | --- | --- |
| | | $\alpha = 1.0$ | $\alpha = 0.1$ |
| ResNet20 | w/o distillation | $86.69 \pm 0.54$ | $70.74 \pm 5.11$ |
| | Basic distillation | $88.63 \pm 0.34$ | $79.98 \pm 2.42$ |
| | Basic distillation w/ warm-up (20 rounds) | $88.66 \pm 0.22$ | $80.20 \pm 1.81$ |
| | Basic distillation w/ warm-up (40 rounds) | $88.47 \pm 0.04$ | $79.27 \pm 2.14$ |
| | Designated distillation | $89.06 \pm 0.19$ | $80.18 \pm 2.38$ |
| ResNet20 (Ensemble) | w/o distillation | $90.58 \pm 0.31$ | $81.76 \pm 2.40$ |
| | Basic distillation | $89.23 \pm 0.36$ | $80.57 \pm 2.24$ |
| | Basic distillation w/ warm-up (20 rounds) | $89.38 \pm 0.42$ | $80.80 \pm 1.95$ |
| | Basic distillation w/ warm-up (40 rounds) | $89.19 \pm 0.24$ | $80.32 \pm 2.55$ |
| | Designated distillation | $90.31 \pm 0.20$ | $81.39 \pm 2.54$ |

Table 3: Comparison between methods. The ratios are relative to FedAvg.

| Method | Communication cost | Ensemble size | No access to client models | Parallelism for server training |
| --- | --- | --- | --- | --- |
| FedAvg | 1.0x | - | ✓ | - |
| FedProx | 1.5x | - | ✓ | - |
| SCAFFOLD | 1.5x | - | ✓ | - |
| FedDF | 1.0x | $C$ | ✗ | ✗ |
| FedBE | 1.0x | $C + S + 1$ | ✗ | ✗ |
| FedEED (w/ FedAvg) | 1.0x | $KR$ | ✓ | ✓ |

$K$: Number of global models    $R$: Number of rounds of time ensemble    $C$: Number of active clients    $S$: Number of sampled models

One possible drawback of the basic scheme is that it may reduce the diversity among the global models and thus hurts the performance of the ensemble, since the global models are forced to produce similar outputs. Thus, we consider another scheme following Codistillation (Anil et al., 2018), where we skip the distillation in the early rounds of training, such that the models will not learn from each other at the beginning to maintain potentially higher diversity.

Lastly, we consider an asymmetric scheme, where only one designated global model will serve as the student and be enhanced by the global trainer. Although there is only one global model being enhanced by distillation in each round of training, the scheme can fit most of the scenarios where only one global model is required for inference, and reduce the cost of distillation. We note that prior work in knowledge distillation (Chen et al., 2020) has applied a similar scheme where a model is chosen to be the main student during distillation, but they did not completely remove the update of the other student models. Compare to their scheme, the all-to-one scheme here is simpler and able to reduce the computation complexity.

In the experiments, we compared the performance of different distillation settings discussed above, including: 1) Simple distillation, where we treat all global model as the student model; 2) Simple distillation with warm-up rounds, which skips the distillation in the early rounds of training; 3) Designated distillation, where we only transfer the knowledge from the ensemble to one designated global model. We use 20 and 40 warm-up rounds for 2).

Results in Table 2 show that, the global model trained by the designated scheme outperformed its counterparts with low degree of Non-IID, and performed as to the others with high degree of Non-IID. On the other hand, the ensembles trained by the designated scheme obtained the best results in all the cases, which is close to the ensemble without distillation. The performance of other ensembles by distillation is much worse than that without distillation. These results support our expectation that asymmetric scheme can help keep the model diversity, and thus improve the performance of both the ensemble and the distilled global model.

## 5 FedEED

In the Sec 4, we showed that two simple schemes: 1) building ensemble from the aggregated global models and 2) designated distillation from the ensemble to the one global model, lead to a competitive instance of the generalized framework, with improved privacy and scalability. We refer the above instance as Federated Efficient Ensemble Distillation (FedEED) and the diagrams of the

ensemble trainer and global trainer are shown in Fig. 1b. FedEED tackled two limitations of the existing distillation-based aggregation methods by utilizing multiple aggregated global models to build the ensemble. Firstly, FedEED does not require direct access to the client models or the model outputs, which is compatible to the mechanism that ensures the users privacy, including secure aggregation (Bonawitz et al., 2017). Secondly, the number of global models is a hyper-parameter, which makes FedEED applicable to large scale FL system with potentially thousands of clients, representing an improved scalability.

Another key advantage of FedEED is the designated distillation scheme. In FedEED, the knowledge of the ensemble will only be transferred to one picked global model with weights $w_{t,0}^{g*}$, such that the diversity between $w_{t,1}^{g*}, ..., w_{t,K-1}^{g*}$ won't be lost after multiple rounds of training. During inference, either the main global model or the ensemble model can be utilized, depending on the device capacity. Besides maintaining the diversity between the models in the ensemble, the designated scheme also allows for parallelism between client-side and server-side computation. In particular, during the computation of $w_{t,0}^{g*}$, other weights $w_{t,1}^{g*}, ..., w_{t,K-1}^{g*}$ are ready to be distributed to the clients for the $t+1$-th round of training, because they will not be enhanced by distillation. This can improve the system throughput by a large factor, especially if the server side training time is not negligible and the clients are not always available. An example of the training pipeline is shown in section A.7 in the appendix.

In summary, the key idea of FedEED is to ensure privacy and scalability, and maintain the diversity between models throughout the whole training process. See Table. 3 for the comparison between FedEED and other methods (McMahan et al., 2017; Li et al., 2020; Karimireddy et al., 2020; Lin et al., 2020; Chen & Chao, 2021).

# 6 EXPERIMENTS: FEDEED

## 6.1 IMAGE CLASSIFICATION WITH CIFAR10/100.

Table 4: CIFAR10/100 results.

| Model | Method | CIFAR10 | | CIFAR100 | |
|---|---|---|---|---|---|
| | | $\alpha = 1.0$ | $\alpha = 0.1$ | $\alpha = 1.0$ | $\alpha = 0.1$ |
| ResNet20 | FedAvg | $88.53 \pm 0.31$ | $78.72 \pm 2.31$ | $58.84 \pm 0.42$ | $52.98 \pm 1.41$ |
| | FedProx | $88.36 \pm 0.18$ | $79.44 \pm 2.17$ | $58.76 \pm 0.93$ | $53.57 \pm 0.37$ |
| | SCAFFOLD | $89.85 \pm 0.50$ | $80.08 \pm 1.53$ | $61.09 \pm 0.46$ | $54.74 \pm 1.26$ |
| | FedDF | $87.98 \pm 0.16$ | $80.04 \pm 1.87$ | $59.74 \pm 0.38$ | $51.83 \pm 0.83$ |
| | FedEED | $89.06 \pm 0.19$ | $80.18 \pm 2.38$ | $61.90 \pm 0.83$ | $54.72 \pm 0.87$ |
| | FedEED ($R = 2$) | $89.01 \pm 0.18$ | $81.27 \pm 2.16$ | $62.42 \pm 0.57$ | $56.39 \pm 1.10$ |
| | FedEED ($R = 4$) | $89.01 \pm 0.07$ | $82.21 \pm 1.10$ | $62.84 \pm 0.45$ | $56.90 \pm 0.77$ |
| ResNet56 | FedAvg | $89.25 \pm 0.11$ | $80.05 \pm 2.65$ | $59.47 \pm 1.23$ | $54.37 \pm 1.24$ |
| | FedProx | $89.65 \pm 0.23$ | $80.84 \pm 1.32$ | $61.01 \pm 1.02$ | $57.18 \pm 0.92$ |
| | SCAFFOLD | $90.73 \pm 0.27$ | $82.22 \pm 0.92$ | $62.94 \pm 0.91$ | $56.36 \pm 0.31$ |
| | FedDF | $88.87 \pm 0.29$ | $81.10 \pm 2.31$ | $60.71 \pm 0.68$ | $52.52 \pm 1.37$ |
| | FedEED | $89.86 \pm 0.14$ | $81.30 \pm 2.17$ | $62.95 \pm 2.43$ | $57.48 \pm 2.03$ |
| | FedEED ($R = 2$) | $90.23 \pm 0.23$ | $82.77 \pm 1.67$ | $65.28 \pm 0.64$ | $59.37 \pm 0.64$ |
| | FedEED ($R = 4$) | $90.00 \pm 0.26$ | $83.20 \pm 0.99$ | $65.63 \pm 0.89$ | $59.56 \pm 0.43$ |
| WRN16-2 | FedAvg | $90.26 \pm 0.33$ | $81.52 \pm 2.66$ | $63.57 \pm 1.06$ | $58.77 \pm 0.55$ |
| | FedProx | $90.38 \pm 0.29$ | $81.78 \pm 2.58$ | $63.75 \pm 0.84$ | $59.24 \pm 1.00$ |
| | SCAFFOLD | $91.32 \pm 0.11$ | $82.90 \pm 2.19$ | $65.33 \pm 0.48$ | $60.66 \pm 0.85$ |
| | FedDF | $89.52 \pm 0.10$ | $81.32 \pm 2.91$ | $63.54 \pm 0.53$ | $56.89 \pm 1.28$ |
| | FedEED | $90.69 \pm 0.41$ | $82.62 \pm 2.46$ | $66.46 \pm 0.30$ | $61.16 \pm 1.20$ |
| | FedEED ($R = 2$) | $90.77 \pm 0.16$ | $83.39 \pm 2.17$ | $67.11 \pm 0.41$ | $62.37 \pm 0.84$ |
| | FedEED ($R = 4$) | $90.70 \pm 0.31$ | $84.02 \pm 1.75$ | $67.48 \pm 0.22$ | $63.05 \pm 0.80$ |

Our experiments of image classification task followed Lin et al. (2020). We trained the models on CIFAR10/100 dataset (Krizhevsky et al., 2009), and utilized CIFAR100 and ImageNet (resized to $32 \times 32$) (Deng et al., 2009) as the unlabelled datasets for the CIFAR10/100 experiments, respectively. There are 100 rounds of training with 20 clients, where 40% of them are active in each round. We sampled data for each of the clients with the Dirichlet distribution (Hsu et al., 2019), with $\alpha = \{1.0, 0.1\}$ for different levels of data heterogeneity. For each setting, we reported the average top-1 accuracy of three runs with different seeds, where the seeds are shared across different methods to generate the same data partition and sequence of client's for a fair comparison. We trained

Table 5: CIFAR10 results of FedEED with different local trainers.

| Model | Method | CIFAR10 (8 clients per round) | | CIFAR10 (20 clients per round) | |
|---|---|---|---|---|---|
| | | $\alpha = 1.0$ | $\alpha = 0.1$ | $\alpha = 1.0$ | $\alpha = 0.1$ |
| ResNet20 | FedAvg | $88.53 \pm 0.31$ | $78.72 \pm 2.31$ | $88.59 \pm 0.28$ | $79.84 \pm 1.50$ |
| | FedProx | $88.36 \pm 0.18$ | $79.44 \pm 2.17$ | $88.52 \pm 0.07$ | $79.63 \pm 2.48$ |
| | SCAFFOLD | $89.85 \pm 0.50$ | $80.08 \pm 1.53$ | $90.10 \pm 0.07$ | $82.83 \pm 2.40$ |
| | FedEED w/ FedAvg | $89.06 \pm 0.19$ | $80.18 \pm 2.38$ | $89.04 \pm 0.11$ | $82.89 \pm 1.30$ |
| | FedEED w/ FedProx | $88.98 \pm 0.22$ | $81.21 \pm 1.30$ | $89.28 \pm 0.17$ | $82.12 \pm 2.26$ |
| | FedEED w/ SCAFFOLD | $88.80 \pm 0.16$ | $76.51 \pm 3.74$ | $90.66 \pm 0.04$ | $84.30 \pm 1.88$ |

ResNet20/56 (He et al., 2016) and WRN16-2 (Zagoruyko & Komodakis, 2016), which are proven capable to achieve high accuracy on CIFAR10/100. For the optimizer, we used SGD with learning rate of 0.8 and 0.1 and batch size of 64 and 256, for the client-side and the server-side training, respectively. We did not apply weight decay or learning rate scheduling in our experiments.

We compared FedEED with FedAvg (McMahan et al., 2017), FedProx(Li et al., 2020), SCAF-FOLD(Karimireddy et al., 2020) and FedDF(Lin et al., 2020). We set the number of local training epoch to be 40, and the regularizer parameter $\mu$ of FedProx to be 0.001. For FedEED, we set the number of global models $K$ to be 4 and used one round of models to build the ensemble, i.e., $R = 1$, unless otherwise specified. For both FedDF and FedEED, we performed distillation with 5000 steps in each training round, with the temperature $\tau$ to be 4.

The results are shown in Table 4. It can be observed that FedEED outperformed all other methods on the harder dataset, CIFAR100, and all settings with high degree of Non-IID. It suggests that FedEED is a good candidate for tackling the Non-IID issue. We also found that the value of $R$ does not affect the performance of FedEED significantly in the CIFAR10 experiments with low degree of Non-IID, but a higher value of $R$ improved the performance of FedEED significantly in the CIFAR10 runs with high degree of Non-IID, and on CIFAR100 datasets with both high and low degree of Non-IID.

## 6.2 COMPATIBILITY OF FEDEED WITH DIFFERENT LOCAL TRAINERS

To evaluate the compatibility of FedEED with different local trainers, we performed experiments by combining FedEED with FedAvg (McMahan et al., 2017) (default option in this paper), FedProx (Li et al., 2020), SCAFFOLD (Karimireddy et al., 2020). The results in Table 5 show that FedEED is compatible with FedProx and the combination achieves better performance with high degree of Non-IID. However, the combination of FedEED and SCAFFOLD did not improve the performance. We expect this is due to the fact that the number of clients per global model is only 2 in FedEED which is too small for SCAFFOLD. We thus performed additional experiments by increasing the ratio of active clients from 0.4 to 1.0, which increases the clients per global model to 5. In these experiments, the combination of FedEED and SCCAFOLD achieved much higher accuracy than the stand-alone FedEED and SCCAFFOLD.

## 6.3 ABLATION STUDY

Ablation study will be provided in the appendix.

## 7 CONCLUSION

In this paper, we studied the key components of distillation-based model aggregation in federated learning. By generalizing the existing methods to a universal framework, we investigated the contribution of different components. It was shown that constructing the ensemble teacher by aggregated models improves the privacy and scalability of the distillation-based federated learning system. Furthermore, the designated distillation, which transfers the knowledge of the ensemble to one global model, helps maintain the diversity among different models used to build the ensemble. Based on these observations, we propose a novel, efficient ensemble distillation method called FedEED, whose performance advantage over existing methods was validated by extensive experiments over the benchmark datasets. The proposed FedEED scheme provides an efficient and scalable way for model aggregation in large-scale learning systems, with improved privacy protection.

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

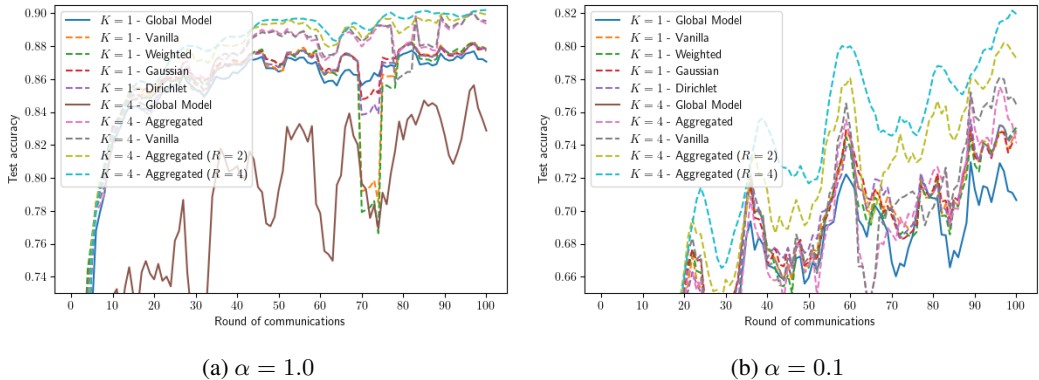

(a) $\alpha = 1.0$        (b) $\alpha = 0.1$

Figure 2: Comparison between different ensemble strategies with CIFAR-10. Note that we applied smoothing to the curves to improve the visibility.

## A APPENDIX

### A.1 ADDITIONAL FIGURES FOR COMPARING DIFFERENT ENSEMBLE STRATEGIES

In this section, we provide the plots of the experiment in section 4.1, as shown in Fig. 2. The settings with 4 global models (i.e. $K = 4$) consistently outperform the others, where the scheme utilizing global models as an ensemble performs closely as that utilizing client models. Utilizing multiple checkpoints of global models as the ensemble obtained the best result in the experiment.

### A.2 FEDEED WITH DIFFERENT COMMUNICATION INTERVALS

Table 6: CIFAR10 results with different communication intervals

| | | | CIFAR10 | |
|---|---|---|---|---|
| # rounds / # local epochs | Model | Method | $\alpha = 1.0$ | $\alpha = 0.1$ |
| 25 / 160 | ResNet20 | FedAvg | $85.81 \pm 0.12$ | $73.17 \pm 3.66$ |
| | | FedDF | $86.78 \pm 0.12$ | $77.79 \pm 1.73$ |
| | | FedEED | $86.27 \pm 0.68$ | $73.99 \pm 2.15$ |
| 100 / 40 | ResNet20 | FedAvg | $88.53 \pm 0.31$ | $78.72 \pm 2.37$ |
| | | FedDF | $87.98 \pm 0.16$ | $80.04 \pm 1.87$ |
| | | FedEED | $89.06 \pm 0.19$ | $80.18 \pm 2.38$ |
| 400 / 10 | ResNet20 | FedAvg | $89.69 \pm 0.27$ | $79.19 \pm 0.42$ |
| | | FedDF | $88.90 \pm 0.19$ | $78.92 \pm 0.59$ |
| | | FedEED | $90.37 \pm 0.08$ | $81.34 \pm 0.81$ |

We compared the performance of FedEED with FedAvg (McMahan et al., 2017) and FedDF (Lin et al., 2020), with difference communication intervals. We fixed the total amount of computation, where we set the number of rounds and the number of local epochs to be 25/100/400 and 160/40/10, respectively. We also scaled the number of distillation steps to 20000/5000/1250. From Table 6, we can observe that FedEED always performed better than FedAvg. But with 25 communication rounds, FedDF outperformed FedEED. We expect this is due to the extremely small number of clients per global models in FedEED, i.e. there are only two clients per global models, which slowed down the convergence in the case with extreme high communication intervals.

### A.3 FEDEED WITH DIFFERENT NUMBER OF GLOBAL MODELS

In the main paper, we used $K = 4$ as our default options of FedEED. Here, we provide results with $K = 2$ to $K = 4$. Note that there are only 8 clients per round in the default configuration, and with $K = 4$ there are only two clients per global model. For $K = 3$, we allocated one more client to the main global model in FedEED. The results in Table 7 show that $K = 4$ is the best configuration of FedEED in the experiment. However, as we mentioned above, there is a trade-off between the speed

Table 7: CIFAR10 results of FedEED with different number of global models.

| | | | CIFAR10 | |
|---|---|---|---|---|
| Model | Method | $K$ | $\alpha = 1.0$ | $\alpha = 0.1$ |
| ResNet20 | FedEED | 2 | $88.34 \pm 0.32$ | $79.38 \pm 2.54$ |
| | | 3 | $88.89 \pm 0.22$ | $79.63 \pm 6.15$ |
| | | 4 | $89.06 \pm 0.19$ | $80.18 \pm 2.38$ |

Table 8: CIFAR10 results of FedEED with different scaling schemes.

| | | | | CIFAR10 | |
|---|---|---|---|---|---|
| # Clients | Model | Method | $K$ | $\alpha = 1.0$ | $\alpha = 0.1$ |
| 8 | ResNet20 | FedAvg | − | $88.53 \pm 0.31$ | $78.72 \pm 2.37$ |
| | | FedDF | − | $87.98 \pm 0.16$ | $80.04 \pm 1.87$ |
| | | FedEED | 4 | $89.06 \pm 0.19$ | $80.18 \pm 2.38$ |
| 14 | ResNet20 | FedAvg | − | $88.67 \pm 0.22$ | $80.25 \pm 1.64$ |
| | | FedDF | − | $88.38 \pm 0.16$ | $81.61 \pm 1.94$ |
| | | FedEED | 4 | $89.14 \pm 0.09$ | $81.26 \pm 1.99$ |
| | | | 7 | $89.31 \pm 0.06$ | $81.90 \pm 1.80$ |
| 20 | ResNet20 | FedAvg | − | $88.59 \pm 0.28$ | $79.84 \pm 1.50$ |
| | | FedDF | − | $88.12 \pm 0.25$ | $81.55 \pm 2.02$ |
| | | FedEED | 4 | $89.04 \pm 0.11$ | $82.89 \pm 1.30$ |
| | | | 10 | $89.14 \pm 0.11$ | $81.97 \pm 1.54$ |

of convergence and the performance of the ensemble teacher, so the optimal value of $K$ may vary with the different tasks.

In Table 8, we further provided the results to show how the performance of FedEED scales with the number of clients, and compared it with FedAvg (McMahan et al., 2017) and FedDF (Lin et al., 2020). As the number of clients increases, we consider two options: 1.) scaling the number of clients per global model, i.e. fixing the value of $K$, 2.) scaling the number of global models, i.e. varying the value of $K$ ($K = 7$ and 10 in Table 8). We found that FedEED outperformed FedAvg and FedDF with both scaling schemes, and obtained the best result when scaling with the number of global models, except the case with high Non-IID with 20 clients. However, in practice, the scaling scheme may be subject to the server capacity, as increasing the number will also increase the cost of distillation.

## A.4 IMPROVEMENT FROM FEDDF

Table 9: The difference between FedDF and FedEED. The method in the last row is identical to the FedEED with $R = 1$.

| | | CIFAR10 | |
|---|---|---|---|
| Model | Method | $\alpha = 1.0$ | $\alpha = 0.1$ |
| ResNet20 | FedDF | $82.53 \pm 0.24$ | $68.72 \pm 4.23$ |
| | + Improved configuration | $87.93 \pm 0.47$ | $79.01 \pm 2.15$ |
| | + Removal of drop-worst & early-stopping | $87.42 \pm 0.19$ | $78.55 \pm 3.00$ |
| | + Aggregated ensemble + $K = 4$ + Designated distillation | $88.35 \pm 0.18$ | $79.59 \pm 2.93$ |

FedEED is similar to FedDF (Lin et al., 2020), because both of them utilize knowledge distillation for model fusion with an unlabelled dataset. In this section, we progressively modify FedDF into FedEED, and show the effect of each of the individual changes. Note that we keep 10% of the training data from CIFAR10 as the validation set of the server in this experiment, following Lin et al. (2020).

The original FedDF utilized Adam optimizer (Kingma & Ba, 2014) for server training, and applied the 'drop-worst' mechanism and the early-stopping. We perform the following modification: 1) adapt the configuration from the main paper (including the use of SGD optimizer), 2) remove the use of 'drop-worst' and early-stopping, and 3) utilize four aggregated global models to build the

ensemble with the designated distillation scheme. By introducing these three modifications, FedDF achieved the same perforamnce as FedEED with $R = 1$.

Table 9 shows the results of different modifications and we have three observations. First, the configuration we used in this paper improved FedDF. Second, the removal of the 'drop-worst' and the early-stopping led to a worse result, but it saved the need of a server validation set. Third, the main mechanisms we used in FedEED, i.e. the ensemble from aggregated model and the designated distillation scheme, provided significant performance improvement.

### A.5   POSSIBLE EXTENSIONS OF FEDEED

To perform distillation, a server dataset or generator is mandatory. In this paper, we consider the use of FedEED with unlabelled server dataset, following Lin et al. (2020); Chen & Chao (2021). Since the dataset is not labelled and can be independent of the client datasets, it is cheap to collect. Note that extending FedEED to labelled dataset or generator is straightforward. We can also extend FedEED to the model heterogeneous setting (Lin et al., 2020; Cho et al., 2022), in which we can pick one global model from each model type as the student models. As long as the number of the picked global models is only a small portion of all models, the diversity among them can be preserved.

### A.6   PRIVACY CONCERN OF FEDEED

Existing distillation based model aggregation methods utilize client models as the teacher for enhancing a global model. For example, FedDF (Lin et al., 2020) is built upon weight averaging based methods like FedAvg (McMahan et al., 2017).

Weight average based methods can benefit from techniques like secure aggregation (Bonawitz et al., 2017), where individual client updates are never disclose to the server and the server can still obtain their weighted sum. However, the above mentioned distillation based methods are not compatible to technique like secure aggregation (Wang et al., 2021). Because the clients are required to send their models or updates to server, and the server will store the individual client models during the process of server-side training, which introduces extra risk as the communication can be intercepted or the server can be attacked.

On the other hand, FedEED does not require access to the raw client models since it utilizes multiple aggregated models as the teacher, which is compatible to the techniques like secure aggregation, where the individual client model updates or weights are never disclosed to the server. This reduces the chance of being attacked during communication or server storage, and achieve the same privacy level as weight average based methods. To summarise, FedEED is able to achieve a higher level of privacy protection than existing distillation based methods.

### A.7   PARALLEL SERVER-SIDE AND CLIENT-SIDE TRAINING OF FEDEED

As mentioned in the main paper, FedEED can introduce parallelism between server-side and client-side training. In this section, we provide an illustration of the training processes for FedEED and other methods (e.g. FedDF (Lin et al., 2020)).

The round time in federated learning depends on the clients' availability. We consider two cases for the clients' availability: 1.) All active clients in a round are able to start local training immediately; 2.) The clients are not always available for training, where we assume that their availability is roughly uniform. The diagram of two training rounds for case 1.) and 2.) are demonstrated in Fig. 3 and Fig. 4, respectively. In these examples, the training time of the clients and the server is assumed constant. We assume a simplified case where there are only 4 clients. For other methods, there are only one global model being updated in each round, where FedEED trains 4 models and always maps the $i$-th model to the $i$-th client. In FedEED, the training of weight $w_{t,k,n}^{l*}$ can be started as soon as all $w_{t-1,k,0}^{l*}, ..., w_{t-1,k,n-1}^{l*}$ are ready and aggregated, for all non-designated models (i.e. $k \neq 0$). However, for simplicity we only consider starting the training of $w_{t,k,0}^{l*}, ..., w_{t,k,n-1}^{l*}$ after all local weight updates in $t - 1$-round have been completed.

In Fig. 3, we demonstrated the parallelism in case 1.). Since FedEED can start some training earlier, the computation of different rounds in FedEED are overlapped, which leads to a longer round time.

However, the average run time of FedEED is the same as the others, which can be defined as the total run time of the training divided the total number of rounds, or the time difference between the completion time of two consecutive rounds. In a more realistic setting, where clients are not always available to start the training, i.e., case 2.), FedEED has much shorter average round time due to the high degree of parallelism, as illustrated in Fig. 4. In FedEED, clients can start their training as long as they are available, where in other methods, clients can not start training before the server-side training is completed.

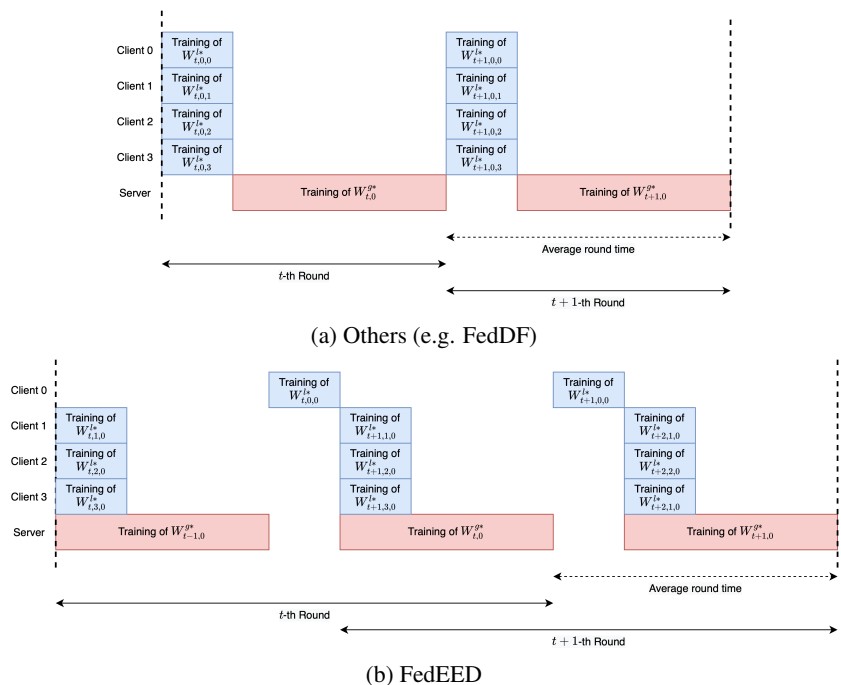

(a) Others (e.g. FedDF)

(b) FedEED

Figure 3: Parallelism comparison between FedEED and other methods, where clients are always available.

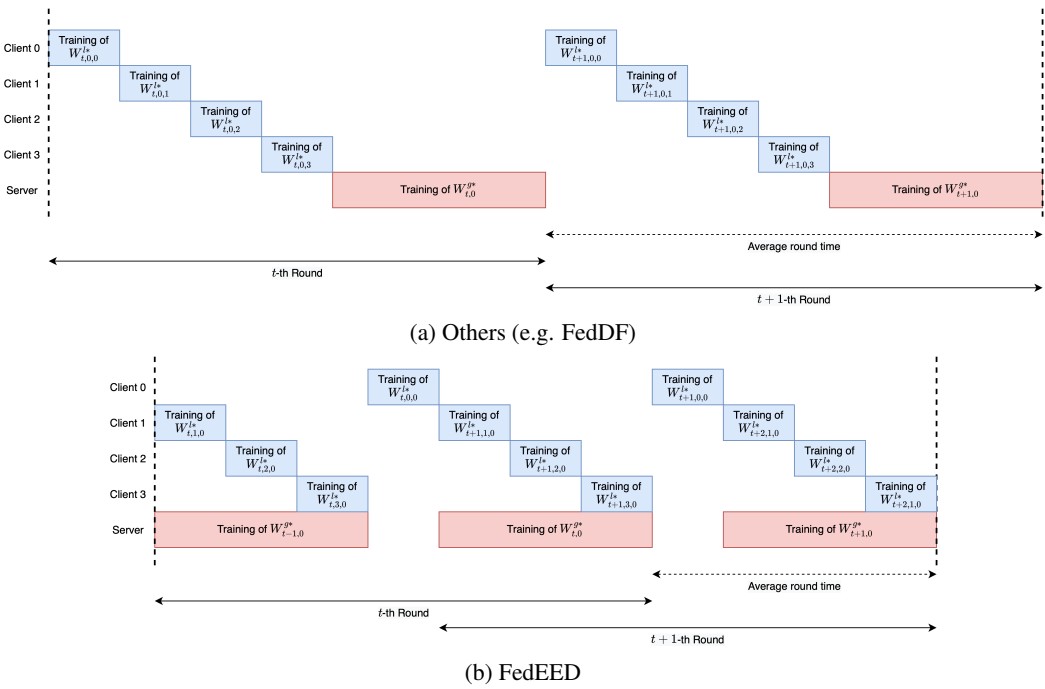

(a) Others (e.g. FedDF)

(b) FedEED

Figure 4: Parallelism comparison between FedEED and other methods, where clients are not always available.

