# OpenReview forum: "FedEED: Efficient Federated Distillation with Ensemble of Aggregated Models"
_ICLR.cc/2023/Conference — Submitted to ICLR 2023_

### Official Review · Reviewer_Cg3f · 2022-10-20

**Confidence:** 3
**Correctness:** 2
**Technical Novelty And Significance:** 1
**Empirical Novelty And Significance:** 2
**Recommendation:** 3

**Clarity, Quality, Novelty And Reproducibility:**

Clarity and quality:
- As stated in the weakness section, I don't think this paper is clear and easy to understand.

================================================================

Novelty:
- The technical novelty is limited. Although this paper summarizes existing works and experiments with several existing methods, it does not present any novel methods or novel ways of applying existing methods.

================================================================

Reproducibility:
Yes if the authors can improve the wording.

**Strength And Weaknesses:**

Strength:
- There is a good effort to summarize the common building blocks across many knowledge-distillation-based FL algorithms. This is helpful for readers to get a high-level overview of algorithms in this class.
- This paper also provides a few popular methods that can be used for each block and performs experiments to compare these methods. The results may be useful when readers try to adapt an existing method to their applications or systems.

Weakness:
- The main issue of this paper is that it is difficult to read and understand. The paper is flooded with undefined terms, symbols, and confusing statements. Also, there is no figure to serve illustration. For example, the paragraph in Sec. 3 states that "In every training round, each of the local trainers will distribute one of the global models to a subset of clients". For FL, the server is typically responsible for distributing models. Does this sentence mean one client will choose a model from the global models and distribute it to other clients, involving something like client-to-client communication? Why does a client store global models in addition to their local models? Another example is that the terms like "global model", "ensemble model", "local trainer", "ensemble trainer", and "global trainer" are not defined, and the symbols in the pseudo code are not defined either.
- There are many unsupported or unexplained claims. For example, Sec. 4.1 has the sentence "Existing distillation-based aggregation methods require all client models to build the ensemble, causing privacy concerns.". Why does this approach cause privacy concerns and why doesn't the proposed approach are not explained nor supported by references. Another example is also from Sec. 4.1. "We noticed the idea of Fed-ensemble, which builds the ensemble from multiple global models for inference purposes. This solves the privacy and scalability issues.". Again, there is no explanation about why this solves the privacy and scalability issues.
- Most of the experiments are on the small CIFAR10 dataset plus the small ResNet20 model except for Table 4, which contains ResNet56 and Cifar100. Because the authors never mention what the metric for the accuracy is, I assume it is top-5 accuracy. ResNet56 on CIFAR100 can easily get a top-5 accuracy above 90%, but the numbers in Table 4 are at the level of 60%. Therefore, I am skeptical about whether the conclusions drawn from these experiments can be generalized to practical settings.
- In Sec. 6.2, there is only one sentence "Ablation study will be provided in the appendix.". The ablation study is an important part of the main paper. Personally, I don't think moving it to the appendix to save space is a good practice. I leave it to the chair to make the judgment.

**Summary Of The Paper:**

This paper investigates applying knowledge distillation to federated learning. It summarizes many related works to give a set of common building blocks and experiments with different choices for each block on ResNet20/56 + Cifar10/100 to see their impact on the accuracy.

**Summary Of The Review:**

Although it is a good effort to summarize related works and experiment with multiple design choices, the technical novelty is somewhat limited. Moreover, this paper needs to be revised to make it easier to understand and better explain and support the claims.

---

> ### Author Response · Authors · 2022-11-18
> **Reply to Reviewer Cg3f**
>
> Thanks for your comment! Following are our responses:
>
> **Comment 1: The main issue of this paper is that it is difficult to read and understand. The paper is flooded with undefined terms, symbols, and confusing statements. Also, there is no figure to serve illustration. For example, the paragraph in Sec. 3 states that "In every training round, each of the local trainers will distribute one of the global models to a subset of clients". For FL, the server is typically responsible for distributing models. Does this sentence mean one client will choose a model from the global models and distribute it to other clients, involving something like client-to-client communication? Why does a client store global models in addition to their local models? Another example is that the terms like "global model", "ensemble model", "local trainer", "ensemble trainer", and "global trainer" are not defined, and the symbols in the pseudo code are not defined either.**
>
> **Response**: Thanks for your comment. We agreed that there were many presentation issues in the previous version. We have significantly revised the paper and included figures for better presentation of the proposed scheme. The terms in the pseudo code have been defined in the description of the algorithm in Sections 3.1 - 3.3.
>
> Sorry for the confusion and we try to explain the key idea here. In this paper, all clients are divided into K groups and each group is taken care of by one local trainer. The global model of one group is the model aggregated from the local models of all clients in that group. The global models from K groups will be utilized to build the ensemble model by the ensemble trainer. Finally, the global trainer will distill the knowledge of the ensemble model to one global model. The global model is distributed from the server to one client, so there is no client-to-client communication.
>
> **Comment 2: There are many unsupported or unexplained claims. For example, Sec. 4.1 has the sentence "Existing distillation-based aggregation methods require all client models to build the ensemble, causing privacy concerns.". Why does this approach cause privacy concerns and why doesn't the proposed approach are not explained nor supported by references. Another example is also from Sec. 4.1. "We noticed the idea of Fed-ensemble, which builds the ensemble from multiple global models for inference purposes. This solves the privacy and scalability issues.". Again, there is no explanation about why this solves the privacy and scalability issues.**
>
> **Response**:  The improved privacy level of FedEED is relative to other distillation-based aggregation methods. Existing methods require the client models to be sent to the server for inference, and the server stores the client models during the distillation steps. This introduces risks during both the communication and distillation steps. On the other hand, FedEED is compatible with techniques like secure aggregation [1], where the server only requires the weighted average of the client models, without accessing the client models. Furthermore, FedEED utilizes aggregated models as the ensemble teacher, and the server does not store the individual client models. Thus, FedEED can prevent the risk that the client-to-server communication is intercepted, or the server is attacked. In the revised manuscript, we have included more details about the privacy concern in the Section A.5 in the appendix.
>
>
> **Comment 3: Most of the experiments are on the small CIFAR10 dataset plus the small ResNet20 model except for Table 4, which contains ResNet56 and CIFAR100. Because the authors never mention what the metric for the accuracy is, I assume it is top-5 accuracy. ResNet56 on CIFAR100 can easily get a top-5 accuracy above 90\%, but the numbers in Table 4 are at the level of 60\%. Therefore, I am skeptical about whether the conclusions drawn from these experiments can be generalized to practical settings.**
>
> **Response**:  In the experiments, the accuracy refers to the top-1 accuracy, which has been explicitly stated in the revised manuscript. The proposed FedEED is able to achieve SOTA performance in the CIFAR datasets. For example, without using weight decay/learning rate decay/momentum, over 90\% top-1 accuracy in CIFAR-10 with Non-IID data is very close to the accuracy in centralized training (without weight decay/learning rate decay/momentum), which has rarely been reported in federated learning research.

---

> > ### Author Response · Authors · 2022-11-18
> > **Reply to Reviewer Cg3f (Continue)**
> >
> > **Comment 4: In Sec. 6.2, there is only one sentence "Ablation study will be provided in the appendix.". The ablation study is an important part of the main paper. Personally, I don't think moving it to the appendix to save space is a good practice. I leave it to the chair to make the judgment.**
> >
> > **Response**: Due to the limited space, we can’t provide additional ablation study in the main paper. However, we studied different components (ensemble trainers, global trainers) before introducing the proposed FedEED algorithm. These experiments serve similar functions as ablation study, as individual components in the framework are studied.
> >
> >
> > **Comment 5: As stated in the weakness section, I don't think this paper is clear and easy to understand.**
> >
> > **Response**: Thanks and we have tried to improve the presentation in the revised manuscript.
> >
> > **Comment 6: The technical novelty is limited. Although this paper summarizes existing works and experiments with several existing methods, it does not present any novel methods or novel ways of applying existing methods.**
> >
> > **Response**: Thanks for your comment. There two major novelties in the proposed FedEED scheme: 1. It trains multiple global models and builds the ensemble from multiple checkpoints of the global models; and 2. it performs designated distillation to enhance only one of the global models to maintain the diversity between different models.
> >
> > For the first novelty, although Fed-ensemble also utilizes multiple global models for inference purposes, the ensemble does not use multiple checkpoints and no distillation is utilized to compress the ensemble into a single model. For the second novelty, no designated scheme has been utilized in federated learning. Besides it ability to keep the diversity among different models, it also leads to reduced computation (no update is required for the non-designated global models) and improved parallelism (parallel client-side training and server-side distillation).
> >
> >
> > [1] Practical Secure Aggregation for Privacy-Preserving Machine Learning

---

### Official Review · Reviewer_KBtv · 2022-10-20

**Confidence:** 5
**Clarity, Quality, Novelty And Reproducibility:** Limited clarity, Limited quality, Lim…
**Correctness:** 2
**Technical Novelty And Significance:** 2
**Empirical Novelty And Significance:** 3
**Recommendation:** 3

**Strength And Weaknesses:**

Strengths:

1. This paper attempts to achieve generalizability of federated distillation, which is a very innovative study.

2. The application of knowledge distillation in the field of federated learning is well researched and summarized.

3. A number of experiments are done using several federated distillation algorithms.

Weaknesses:

1. In abstract
	- This paper first proposes a generalized distillation framework which divides the federated distillation process into three key stages. However, there are various types of federated distillation, and the three-stage framework is only one of them, so it should not be called a generalized distillation framework.
	- FedEED also achieves a higher level of privacy protection, because the access to client models is no longer required. However, The client model still needs to be transferred between local trainer and  ensemble trainer, which means the client model can still be attacked.

2. In framework
	- This paper describes in detail the three-stage process of the generalized federated distillation framework, wouldn't it be better to add a framework figure to illustrate it more visually?
	- This framework includes K local trainers, an ensemble trainer, and a global trainer, but traditional federated learning includes K clients and a server, what is the correspondence between them, this paper does not explain.

3. In experiments
	- This paper mentions that the number of clients is 20, with 40% of activated clients, which means that there are very few clients, which is contrary to this paper's emphasis on the fact that FedEED achieves improved scalability in large-scale systems. Therefore, experiments on more clients should be conducted to demonstrate the scalability of FedEED.
	- The correspondence between Figure 1 and Table 1 is not clear, and the introduction part is also a bit messy, which is easy to cause confusion. If you follow the order in Figure 1, the order in Table 1 would be “Global model(K=1)、Ensemble(K=1,Clients)、Ensemble(K=1,Clients,Weighted)、Ensemble(K=1,Bayesian,Gaussian)、Ensemble(K=1,Bayesian,Dirichlet)、Global model(K=4)、Ensemble(K=4,Clients)、Ensemble(K=4,Aggregated)、Ensemble(K=4,R=2,Aggregated)、Ensemble(K=4,R=4,Aggregated)”.
	- The dataset CIFAR100 is included in the description of Table 2, but it is not available in the experimental results. In addition, only Table 4 conducted experiments on CIFAR100, why not include this dataset in other experiments.

**Summary Of The Paper:**

This paper first propose a generalized distillation framework. Based on the study of each component in the generalized framework, this paper proposes Federated Efficient Ensemble Distillation(FedEED), which is efficient, scalable and privacy.

**Summary Of The Review:**

This paper presents an innovative research of ederated distillation, But I have doubts about the scalability and privacy of the proposed framework FedEED.

---

> ### Author Response · Authors · 2022-11-18
> **Reply to Reviewer KBtv**
>
> Thanks for your comment! Following are our responses:
>
> **Comment 1: This paper first proposes a generalized distillation framework which divides the federated distillation process into three key stages. However, there are various types of federated distillation, and the three-stage framework is only one of them, so it should not be called a generalized distillation framework.**
>
> **Response**: Thanks for the comments. We have revised the paper and refer the framework as the ‘generalized framework for distillation-based model aggregation’, which we believe can better represent the works that utilized distillation in a similar manner.
>
>
> **Comment 2: FedEED also achieves a higher level of privacy protection, because the access to client models is no longer required. However, The client model still needs to be transferred between local trainer and ensemble trainer, which means the client model can still be attacked.**
>
> **Response**: The improved privacy level of FedEED is relative to other distillation-based aggregation methods. Existing methods require the client models to be sent to the server for inference, and the server stores the client models during the distillation steps. This introduces risks during both the communication and distillation steps. On the other hand, FedEED is compatible with techniques like secure aggregation [1], where the server only requires the weighted average of the client models, without accessing the client models. Furthermore, FedEED utilizes aggregated models as the ensemble teacher, and the server does not store the individual client models. Thus, FedEED can prevent the risk that the client-to-server communication is intercepted, or the server is attacked. In the revised manuscript, we have included more details about the privacy concern in the Section A.5 in the appendix.
>
>
> **Comment 3: This paper describes in detail the three-stage process of the generalized federated distillation framework, wouldn't it be better to add a framework figure to illustrate it more visually?**
>
> **Response**: Thanks for the comments. We have added Fig. 1 to illustrate the structure of the generalized framework and the proposed FedEED.
>
>
> **Comment 4: This framework includes K local trainers, an ensemble trainer, and a global trainer, but traditional federated learning includes K clients and a server, what is the correspondence between them, this paper does not explain.**
>
> **Response**: The proposed framework has the same physical network as the traditional federated learning systems. In particular, there are certain amount of clients and one server. For ease of illustration, let's assume there are C clients (Note that C is the second K mentioned by the reviewer). The proposed framework first divide the C clients into K (this is the first K mentioned by the reviewer) groups and each group is taken care of by one local trainer. So, the number K in this paper is not the number of clients, but the number of groups or the number of local trainers.
>
> **Comment 5: This paper mentions that the number of clients is 20, with 40\% of activated clients, which means that there are very few clients, which is contrary to this paper's emphasis on the fact that FedEED achieves improved scalability in large-scale systems. Therefore, experiments on more clients should be conducted to demonstrate the scalability of FedEED.**
>
> **Response**: Due to the limited computational resources, we could not simulate a real large scale system, especially if we need to compare to existing methods that have a complexity scale with the number of clients. Compared with existing methods that use all client models to build the ensemble, the proposed FedEED divides the clients into K groups and trains K global models to construct the ensemble. When the number of clients in a federated learning system increases, FedEED can keep the same number of global models by increasing the number of clients in each group. As a result, the computation complexity of the ensemble trainer will not increase with the network size. Furthermore, with the designated distillation, only one global model will be updated in each round of training. Thus, the complexity of the global trainer also will not increase with the network size.
>
> Furthermore, as shown in Section A.3, when the number of clients in each training round increases, the size of the ensemble for FedDF increases, but for FedEED with K=4, the ensemble size is a constant. When there are 20 clients, FedEED still outperforms FedDF despite having a five time smaller ensemble. In our reply to reviewer Ybra, we provided the round time comparison, which show that FedDF scale better than FedDF, a representing work in distillation based aggregation methods.

---

> > ### Author Response · Authors · 2022-11-18
> > **Reply to Reviewer  KBtv (Continue)**
> >
> > **Comment 6: The correspondence between Figure 1 and Table 1 is not clear, and the introduction part is also a bit messy, which is easy to cause confusion. If you follow the order in Figure 1, the order in Table 1 would be “Global model(K=1)、Ensemble(K=1,Clients)、Ensemble(K=1,Clients,Weighted)、Ensemble(K=1,Bayesian,Gaussian)、Ensemble(K=1,Bayesian,Dirichlet)、Global model(K=4)、Ensemble(K=4,Clients)、Ensemble(K=4,Aggregated)、Ensemble(K=4,R=2,Aggregated)、Ensemble(K=4,R=4,Aggregated)”.**
> >
> > **Response**: We have changed the order in the table for better presentation. The table also matches the figures now (the figures have been moved to appendix).
> >
> >
> > **Comment 7: The dataset CIFAR100 is included in the description of Table 2, but it is not available in the experimental results. In addition, only Table 4 conducted experiments on CIFAR100, why not include this dataset in other experiments.**
> >
> > **Response**:  For Table 2, we made a mistake and the experiment in that section did not not include CIFAR100.
> >
> > For Table 4, we investigated different options for ensemble trainer and global trainer on the CIFAR10 dataset. Then, we compare the trained FedEED with other methods on both CIFAR10 and CIFAR100 datasets. It is shown that, even the FedEED was trained on CIFAR10 dataset, it consistently outperforms all other methods on CIFAR100 dataset. This is the reason why CIFAR100 only occurs in Table 4.
> >
> >
> > **Comment 8: This paper presents an innovative research of federated distillation, But I have doubts about the scalability and privacy of the proposed framework FedEED.**
> >
> > **Response**: Thanks for your comments. In the following, we highlight the innovation in terms scalability and privacy.
> >
> > 1. Scalability
> > Compared with existing methods that use all client models to build the ensemble, the proposed FedEED divides the clients into K groups and trains K global models to construct the ensemble. When the number of clients in a federated learning system increases, FedEED can keep the same number of global models by increasing the number of clients in each group. As a result, the computation complexity of the ensemble trainer will not increase with the network size. Furthermore, with the designated distillation, only one global model will be updated in each round of training. Thus, the complexity of the global trainer also will not increase with the network size.
> >
> > 2. Privacy
> > The improved privacy level of FedEED is relative to other distillation-based aggregation methods. Existing methods require the client models to be sent to the server for inference, and the server stores the client models during the distillation steps. This introduces risks during both the communication and distillation steps. On the other hand, FedEED is compatible with techniques like secure aggregation [1], where the server only requires the weighted average of the client models, without accessing the client models. Furthermore, FedEED utilizes aggregated models as the ensemble teacher, and the server does not store the individual client models. Thus, FedEED can prevent the risk that the client-to-server communication is intercepted, or the server is attacked. In the revised manuscript, we have included more details about the privacy concern in the Section A.5 in the appendix.
> >
> > [1] Practical Secure Aggregation for Privacy-Preserving Machine Learning

---

### Official Review · Reviewer_4iWR · 2022-10-23

**Confidence:** 4
**Correctness:** 3
**Technical Novelty And Significance:** 2
**Empirical Novelty And Significance:** 2
**Recommendation:** 5

**Clarity, Quality, Novelty And Reproducibility:**

The problem this paper address is clear. The logic of the article is clear. Somewhat novelty. Repeatable in a limited simulation scenario.

**Strength And Weaknesses:**

Strength:
Due to the use of aggregated models, FedEED also achieves a higher level of privacy protection. In addition, the knowledge distillation in FedEED only happens from the ensemble teacher to a designated model such that the diversity among different aggregated models is maintained.

Weaknesses:
The authors evaluated the performance of the proposed method only on the task of Non-IID labels and did not validate the performance of the proposed method on the Non-IID feature task, which is a more realistic scenario because there is often a large domain gap problem in the distribution of data features of different clients as the different data acquisition devices, environments, and qualities. Therefore it does not provide a comprehensive assessment of the generality of the proposed method.

**Summary Of The Paper:**

In this paper, the authors studied the key components of distillation-based model aggregation in FL. Different from existing approaches, the ensemble teacher of FedEED is constructed by aggregated models, instead of client models, to achieve improved scalability in large-scale systems.

**Summary Of The Review:**

1.  The authors should add experiments to further validate the performance of the proposed method on Non-IID feature tasks to further evaluate the generality of the proposed method.
2. The authors' presentation is not clear and the architecture of the algorithm cannot be well understood. Perhaps the authors can give a clear architecture diagram to assist the readers to understand the essence of the proposed method clearly.
3. Several general FL methods have been proposed and achieved excellent performance, such as FedBN, FedRep, and FedDC, and the authors should analyze the performance of more SOTA FL methods to further evaluate the proposed approach.

a) Li X, Jiang M, Zhang X, et al. Fedbn: Federated learning on non-iid features via local batch normalization. arXiv preprint arXiv:2102.07623, 2021.
b) Collins L, Hassani H, Mokhtari A, et al. Exploiting shared representations for personalized federated learning. International Conference on Machine Learning. PMLR, 2021: 2089-2099.
c) Gao L, Fu H, Li L, et al. FedDC: Federated Learning with Non-IID Data via Local Drift Decoupling and Correction. Proceedings of the IEEE/CVF Conference on Computer Vision and Pattern Recognition. 2022: 10112-10121.

---

> ### Author Response · Authors · 2022-11-18
> **Reply to Reviewer 4iWR**
>
> Thanks for your comment! Following are our responses:
>
> **Comment 1: The authors evaluated the performance of the proposed method only on the task of Non-IID labels and did not validate the performance of the proposed method on the Non-IID feature task, which is a more realistic scenario because there is often a large domain gap problem in the distribution of data features of different clients as the different data acquisition devices, environments, and qualities. The authors should add experiments to further validate the performance of the proposed method on Non-IID feature tasks to further evaluate the generality of the proposed method.**
>
> **Response**: We agree with the reviewer that federated feature learning is an important and realistic task. However, as it is not yet a common task in comparing federated learning algorithms, we did not include it in our research. But we expect FedEED to be compatible with such tasks, by replacing the corresponding loss function (e.g. from KL-divergence to L2 loss), since knowledge distillation has also been utilized for feature learning.
>
> **Comment 2: The authors' presentation is not clear and the architecture of the algorithm cannot be well understood. Perhaps the authors can give a clear architecture diagram to assist the readers to understand the essence of the proposed method clearly.**
>
> **Response**: Thanks for the comments. We are sorry that the presentation in the previous version is not clear. We have revised the paper and included the architecture diagram for better presentation of the proposed scheme. Please see Section 3 and Fig. 1 for more details.
>
>
> **Comment 3: Several general FL methods have been proposed and achieved excellent performance, such as FedBN, FedRep, and FedDC, and the authors should analyze the performance of more SOTA FL methods to further evaluate the proposed approach.**
>
> **Response**: We agree it is better to include more comparisons. Unfortunately, there are different settings (e.g. the degree and the implementation of N.I.I.D, the number of clients, the number of rounds) in different papers for federated learning systems. We will need to run all methods to perform fair comparison. In this work, we focused on distillation-based FL and tried to compare several state-of-the-art distillation-based schemes, so we only focus on comparing with some representing works. Due to the large number of experiments, we are not able to add additional baselines to the paper.

---

### Official Review · Reviewer_Ybra · 2022-10-24

**Confidence:** 5
**Correctness:** 2
**Technical Novelty And Significance:** 1
**Empirical Novelty And Significance:** 2
**Recommendation:** 3

**Clarity, Quality, Novelty And Reproducibility:**

Quality: generally good.
Clarity: clear written except some claims about scalability, privacy, efficiency. (see weakness)
Originality: limited novelty, even the group-division ensemble is not new (Fed-ensemble proposes similar group division method). Could you explain the difference with Fed-ensemble?

**Strength And Weaknesses:**

Strength:
1. The proposed group-division ensemble framework can be generalized to different existing FL methods. First, aggregation inside individual group helps to improve the level of parallel.
2. Experiments show the proposed methods achieves better performance than directly aggregating all locals.

Weakness:
1. Limited novelty: the authors only replace the typical aggregation in FL with two stages: group aggregation and then global aggregation. The detailed ensemble strategies (e.g. bayesian ensemble) are borrowed from existing works. The novelty and contribution is limited considering the method is just simple combination.
2. Overclaim about scalability: I cannot see any improvement on scalability from the proposed method and experiments design.
3. Overclaim about privacy: In abstract it says "FedEED also achieves higher level of privacy protection, because the access to client models is no longer required." But the method still iteratively upload local model parameters besides the group global model. [1][2] demonstrated that local private data could be fully recovered from publicly-shared gradients (iteratively shared local model parameters).
3. Overclaim about efficiency: No quantitative analysis of efficiency metrics such as latency, communication bandwidth. Considering the "ensemble trainer" also involves the participation of local models, I doubt if the latency and communication bandwidth could be even higher than the existing FL methods for comparison. It is meaningless to show #rounds in the table, as the latency and bandwidth of each round in your method is obviously higher than other FL methods. The communication cost in Table3 is very ambiguous without any analysis or calculation.
4. Experiments: only have experiments on cifar10/cifar100. Should do more through experiments under typical FL setting: such as EMNIST in SCAFFOLD or NLP tasks (AG News, SST2 in FedDF).

[1] Deep leakage from gradients.
[2] Inverting gradients–how easy is it to break privacy in federated learning?

**Summary Of The Paper:**

The paper proposes a generalized federated distillation framework which divides local nodes into groups: the aggregation is first inside each group and then from groups into the final global model.  The advantage is groups can do each aggregation in parallel in the first stage.
The experiments exhaust several popular federated distillation method for the aggregation. The results show the proposed group-division ensemble strategy does improve over the ensemble directly from all locals.

**Summary Of The Review:**

Although the experiments show better performance over the baseline methods, the novelty is very limited. And the efficiency metrics are not clear. I also doubt the claims regarding privacy and scalability.   Since the "efficiency" is the main point of the paper, I hope the authors could provide more efficiency analysis with quantitative results on latency, communication bandwidth.

---

> ### Author Response · Authors · 2022-11-18
> **Reply to Reviewer Ybra**
>
> Thanks for your comment! Following are our responses:
>
> **Comment 1: Limited novelty: the authors only replace the typical aggregation in FL with two stages: group aggregation and then global aggregation. The detailed ensemble strategies (e.g. bayesian ensemble) are borrowed from existing works. The novelty and contribution is limited considering the method is just simple combination.**
>
> **Response**: Thanks for your comment. There two major novelties in the proposed FedEED scheme: 1. It trains multiple global models and builds the ensemble from multiple checkpoints of the global models; and 2. it performs designated distillation to enhance only one of the global models to maintain the diversity between different models.
>
> For the first novelty, although Fed-ensemble also utilizes multiple global models for inference purposes, the ensemble does not use multiple checkpoints and no distillation is utilized to compress the ensemble into a single model. For the second novelty, no designated scheme has been utilized in federated learning. Besides it ability to keep the diversity among different models, it also leads to reduced computation (no update is required for the non-designated global models) and improved parallelism (parallel client-side training and server-side distillation).
>
>
> **Comment 2: Overclaim about scalability: I cannot see any improvement on scalability from the proposed method and experiments design.**
>
> **Response**: Compared with existing methods that use all client models to build the ensemble, the proposed FedEED divides the clients into K groups and trains K global models to construct the ensemble. When the number of clients in a federated learning system increases, FedEED can keep the same number of global models by increasing the number of clients in each group. As a result, the computation complexity of the ensemble trainer will not increase with the network size. Furthermore, with the designated distillation, only one global model will be updated in each round of training. Thus, the complexity of the global trainer also will not increase with the network size.
>
> We can provide some numbers regarding the scaling of FedDF and FedEED, but we didn't include them in the paper as we believe that these numbers does not reflect the real situation: our experiments were running on a single gpu system, where in real federated system, the client-side training is distributed, and the server-side training is happened on the central server. It is not clear about the ratio between the computation resources of the client-side and the server-side, so the experiments on a single gpu system can only be consider as reference.
>
> Below are the round time (in seconds) in our experiment on an A100 GPU, with the configuration in the paper, except we scale the number of active clients.
>
> ResNet20:
>
> |        |        | Number of active clients |        |
> |--------|--------|--------------------------|--------|
> |        | 8      | 14                       | 20     |
> | FedAvg | 122.91 | 213.89                   | 304.01 |
> | FedDF  | 298.51 | 460.78                   | 619.62 |
> | FedEED | 297.86 | 387.26                   | 482.95 |
>
> WResNet16-2:
>
> |        |        | Number of active clients |        |
> |--------|--------|--------------------------|--------|
> |        | 8      | 14                       | 20     |
> | FedAvg | 123.63 | 214.89                   | 301.62 |
> | FedDF  | 347.21 | 551.84                   | 748.72 |
> | FedEED | 297.70 | 391.99                   | 477.16 |
>
> We can also estimate the round time of FedDF and FedEED that has been spending on distillation, by computing the difference of their round time to the time of FedAvg. E.g., with WResNet-16-2 and 8 clients, FedDF spent ~223s where FedEED only spent ~174s, which FedDF spent 1.28x times for performing distillation. When increasing the number of clients to 20, FedDF spent ~447s where FedEED only spent ~175s, which FedDF spent 2.55x times for distillation. We also found that the above experiment does not fully utilize the GPU, when training larger models, the difference between FedDF and FedEED will be even larger. For example, with WResNet16-8, the round time of FedAvg/FedDF/FedEED are 555.81/2625.36/1202.69, which the distillation time of FedDF is 3.20x of FedEED.
>
> Furthermore, above experiment does not consider the parallelism of FedEED. If the distillation time is short than the local training time, it is possible to hide the distillation time in real world system, e.g. when there are 20 clients in the above case with WResNet16-2, the local training time is ~302s, where the distillation time of FedEED is ~176s.

---

> > ### Author Response · Authors · 2022-11-18
> > **Reply to Reviewer Ybra (Continue)**
> >
> > **Comment 3:  Overclaim about privacy: In abstract it says "FedEED also achieves higher level of privacy protection, because the access to client models is no longer required." But the method still iteratively upload local model parameters besides the group global model. [1][2] demonstrated that local private data could be fully recovered from publicly-shared gradients (iteratively shared local model parameters).**
> >
> > **Response**:  First of all, we want to clarify that, although the generalized framework has the option to upload local model parameters, the proposed FedEED scheme only requires the global models, which can be securely aggregated from client models.
> >
> > The improved privacy level of FedEED is relative to other distillation-based aggregation methods. Existing methods require the client models to be sent to the server for inference, and the server stores the client models during the distillation steps. This introduces risks during both the communication and distillation steps. On the other hand, FedEED is compatible with techniques like secure aggregation [1], where the server only requires the weighted average of the client models, without accessing the client models. Furthermore, FedEED utilizes aggregated models as the ensemble teacher, and the server does not store the individual client models. Thus, FedEED can prevent the risk that the client-to-server communication is intercepted, or the server is attacked. In the revised manuscript, we have included more details about the privacy concern in the Section A.5 in the appendix.
> >
> >
> > **Comment 4:  Overclaim about efficiency: No quantitative analysis of efficiency metrics such as latency, communication bandwidth. Considering the "ensemble trainer" also involves the participation of local models, I doubt if the latency and communication bandwidth could be even higher than the existing FL methods for comparison. It is meaningless to show the number of rounds in the table, as the latency and bandwidth of each round in your method is obviously higher than other FL methods. The communication cost in Table3 is very ambiguous without any analysis or calculation.**
> >
> > **Response**: The proposed FedEED has the same communication workload as existing distillation-based method like FedDF. But, FedEED has two advantages with respect to computation efficiency. First, as shown in Section A.3, when the number of clients in each training round increases, the size of the ensemble for FedDF increases, but for FedEED with K=4, the ensemble size is a constant. When there are 20 clients, FedEED still outperforms FedDF despite having a five time smaller ensemble. Second, as mentioned in Section 5 in the paper and Section A.6 in the appendix, the designated distillation enables parallel computation between the client-side and the server-side.
> >
> >
> > **Comment 5:  Experiments: only have experiments on cifar10/cifar100. Should do more through experiments under typical FL setting: such as EMNIST in SCAFFOLD or NLP tasks (AG News, SST2 in FedDF).**
> >
> > **Response**: We are willing to provide more experiment results with additional dataset. However, due to the limited time of the rebuttal period, we have not been able to complete the experiments yet. We will provide the results for the EMNIST and NLP tasks in the camera-ready version or later version of the appendix.
> >
> > **Comment 6:  Could you explain the difference with Fed-ensemble?**
> >
> > **Response**: As mentioned in the reply to Comments 1, Fed-ensemble utilizes multiple global models for inference but not training purposes. Furthermore, the ensemble of Fed-ensemble does not use multiple checkpoints and no distillation is utilized to compress the ensemble into a single model.
> >
> >
> > [1] Practical Secure Aggregation for Privacy-Preserving Machine Learning
> > [2] A Field Guide to Federated Optimization

---

### Decision · Program_Chairs · 2023-01-20

**Decision:**

Reject

**Justification For Why Not Higher Score:**

* Lack of novelty (dividing the aggregation into stages is not particularly novel)
* Unsupported claim that the proposed technique offers better privacy than other federated learning techniques
* Unsupported claim of improved efficiency (no timing results)
* Unclear manuscript

**Justification For Why Not Lower Score:**

NA

**Metareview: Summary, Strengths And Weaknesses:**

The paper proposes a new federated learning technique that does the aggregation of client models via knowledge distillation in stages.

Strength:
* Nice survey of knowledge distillation techniques for federated learning
* Extensive experiments with many knowledge distillation techniques for federated learning

Weaknesses:
* Lack of novelty (dividing the aggregation into stages is not particularly novel)
* Unsupported claim that the proposed technique offers better privacy than other federated learning techniques
* Unsupported claim of improved efficiency (no timing results)
* Unclear manuscript

Unfortunately, this work does not have enough novelty for publication and it suffers from unsupported claims as well as a lack of clarity.